# Tuning the Nucleophilicity and Electrophilicity of Group 10 Elements through Substituent Effects: A DFT Study

**DOI:** 10.3390/ijms242115597

**Published:** 2023-10-26

**Authors:** Sergi Burguera, Antonio Bauzá, Antonio Frontera

**Affiliations:** Department of Chemistry, Universitat de les Illes Balears, Ctra. de Valldemossa Km 7.5, 07122 Palma, Baleares, Spain; sergi.burguera@uib.es (S.B.); antonio.bauza@uib.es (A.B.)

**Keywords:** supramolecular chemistry, non-covalent interactions, group 10, Ni, Pd, Pt, crystal engineering, electron donor, electron acceptor, theoretical chemistry

## Abstract

In this study, a series of electron donor (–NH_2_, –NMe_2_ and –^t^Bu) and electron-withdrawing substituents (–F, –CN and –NO_2_) were used to tune the nucleophilicity or electrophilicity of a series of square planar Ni^2+^, Pd^2+^ and Pt^2+^ malonate coordination complexes towards a pentafluoroiodobenzene and a pyridine molecule. In addition, Bader’s theory of atoms in molecules (AIM), noncovalent interaction plot (NCIplot), molecular electrostatic potential (MEP) surface and natural bond orbital (NBO) analyses at the PBE0-D3/def2-TZVP level of theory were carried out to characterize and discriminate the role of the metal atom in the noncovalent complexes studied herein. We hope that the results reported herein may serve to expand the current knowledge regarding these metals in the fields of crystal engineering and supramolecular chemistry.

## 1. Introduction

Supramolecular chemistry is the tip of the spear in modern day chemistry, providing essential tools for the fields of medicinal chemistry [1,2], materials science [3,4,5], crystal engineering [6,7] and catalysis [8,9]. Thus, a comprehensive and thorough study of the non-covalent interactions that guide and direct the formation of these supramolecular assemblies is imperative.

Group 10 elements Ni, Pd and Pt have been extensively exploited for some crucial applications, mainly in catalytic reactions [10,11,12,13]. Moreover, Pt remains the fundamental keystone for antitumoral drugs [14,15,16,17]. However, the supramolecular chemistry of these elements has not yet been intensively studied to its full potential and it could prove to be useful in the different aforementioned fields.

Recently, we studied the possible capacity of these group 10 elements to act as electron donors, having a π-acidic molecule acting as electron acceptor [18,19,20,21], based on crystal structures deposited in the Cambridge Structural Database (CSD) [22]. This interaction could be explained due to the π-hole nature of the acceptor and a filled d_z2_ orbital of the metal in square planar complexes. It has also been reported that these metals can act as electron donors in different σ-hole interactions [23,24,25,26,27].

It is well known that the behavior of different chemical centers can be modified depending on the influence of the substituents at immediate proximities or at more long-range distances due to the inductive and resonant effects of the substituents.

In this work, we designed several exemplary complexes involving Ni^II^, Pd^II^ and Pt^II^ in a square planar geometry, and we investigated their ability to act as electron acceptors (against a pyridine ring) or as electron donors (against a halogen σ-hole-containing moiety, pentafluoroiodobenzene). We also studied how their capacity to act in either one of those roles would be influenced by short- and long-range effects This was accomplished using different substituents with electron-donating or electron-accepting capacities. Thus, this study could present a way of deepening our knowledge on the supramolecular behavior and capabilities of the group 10 elements. We focused on this group of elements due to its universal relevance and because of their ability to act as both electron acceptors and electron donors thanks to having some high-energy d orbitals. The results herein obtained demonstrate that the acceptor/donor capacity of these metals can be finely tuned by the immediate or distant substituents, proving that these metals can be further exploited in supramolecular chemistry.

## 2. Results and Discussion

Several systems containing different vicinal and distant substituents were designed and examined. These systems were designed to contain one of the three studied metal cations (Ni, Pd or Pt) in the +2 oxidation state and in a square planar geometry with a 0 formal charge for the systems. They were complexed by oxygen or sulfur atoms to account for the short-range effects’ variation. To account for the long-range effects, these monomers were also modified using different behavioral substituents with commonly known donor/acceptor properties. The proposed substituents were: -H, -F, -CN, -NO_2_, -NH_2_, -N(CH_3_)_2_ and the tert-butyl group -C(CH_3_)_3_. The proposed monomers are summarized and numerated in Figure 1. These structures were proposed in order to study a single center (Ni, Pd or Pt) acting as both donor/acceptor in posterior analyses.

### 2.1. Molecular Electrostatic Potential Surface Analysis (MEP)

The optimized ML_2_ complexes (see Section 3) were then analyzed by means of the MEP analysis. This method of analysis is a useful tool for identifying regions of highly positive and/or negative electrostatic potential. This gives a considerable amount of valuable information for either predicting, explaining or rationally designing efficient compounds which can have a major impact in various fields. Some recent, relevant examples include the design of high-performance zinc batteries [28,29] and the interesting study of the terminal σ-hole in ultrathin nanorods which directs and enhances the adsorption at these terminal ends [30]. For each monomer, the maximum, minimum and over the metal potential values are displayed in Table 1. Some exemplary MEPs are shown in Figure 1. The rest are displayed in Appendix A.

The maximum electrostatic potential of the monomers (V_s, max_) ranged from +13.8 to +39.1 kcal/mol, obtained from compounds **8B** (M = Pd^II^, Y = S, X = -H) and **19B** (M = Pt^II^, Y = S, X = -NH_2_), respectively. These maximum electrostatic potentials were located in the vast majority of cases at the H- in the molecular plane, in between the X- and Y- groups (see Figure 1).

The minimum electrostatic potential of the monomers (V_s, min_) ranged from −40.8 to −9.7 kcal/mol, from the compounds **6A** (M = Ni^II^, Y = O, X = -N(CH_3_)_2_) and **9B** (M = Pd^II^, Y = S, X = -F). These minima were in most cases located in the molecular plane, on the Y-M^II^-Y bisectors looking to the outside of the molecule (see Figure 1). It is also worth noting that the A set of compounds (Y = O) had an average of around −35 kcal/mol for the minimum electrostatic potential. However, the B set of compounds (Y = S) had an average of around −20 kcal/mol. The sulfur soft center dissipated the minimum electrostatic potential of the monomers even to the point where, in some cases, the minimum was close to or located over the metal center, perpendicular to the molecular plane (monomers **8B**, **13B**, **14B**, **15B**, **16B**, **20B** and **21B**). These particular cases where the electrostatic potential over the metal center was extremely close to the minimum, or indeed was the minimum, could be explained considering the nature of the Y and X substituents, as well as the increasing nucleophilicity of the metal when descending in group 10 of the elements. For instance, all these cases were from the B set of compounds (Y = S), four of them with the same electron-donor moieties (X = -N(CH_3_)_2_, -tBu) and with the metal center being a Pd^II^ or a Pt^II^, the latter containing the V_s, min_ above it, while the former’s potential over the center was similar to the V_s, min_.

Although some trends can be appreciated when comparing the electrostatic potential maxima and minima of the different structures, the strongest conclusions can be drawn when comparing the electrostatic potential at the same place. Thus, it is interesting to analyze and compare the electrostatic potential above the metal centers in all the compounds.

The best way to compare these monomers and the effects of the different variables is to fix two of our three sets of variables and observe the trend. Take, for instance, compounds **1A**, **8A** and **15A** (thus, having fixed X = -H and Y = O). We can notice that the potential over the metal becomes more negative when going deeper in the metal group, as expected. That is, for X, Y = constant: V_s, Ni_^II^ > V_s, Pd_^II^ > V_s, Pt_^II^. That is true for all these possible 14 sets where the only variable is the metal center.

Subsequently, for X, M^II^ = constant (and therefore having Y as the variable), the V_s, M_^II^ _(Y = O)_ values are greater than the V_s, M_^II^ _(Y = S)_ ones. This could be explained due to the hard electron-withdrawing capability of the oxygen atoms when compared to the softer sulfur centers. Interestingly, for each of these sets, where also the metal remained constant, it can be noticed that the diminution of electrostatic potential over the metal center caused by the exchange of oxygen atoms with sulfur atoms is roughly about 8 kcal/mol for Ni^II^, 6 kcal/mol for Pd^II^ and 3 kcal/mol for the Pt^II^ sets, slightly varying from these values depending on the X substituent.

Finally, when comparing sets where the metal center and the chalcogen atom are unaltered, that is, Y, M^II^ = constant, the following trend can be extracted: V_s, M_^II^ X = (-NO_2_ > -CN >> -F >> -H > -NH_2_ > -tBu > N(CH_3_)_2_). Furthermore, analyzing the six possible sets, the amount of electrostatic potential variation when switching between the different substituents can be roughly estimated. For example, when changing the -NO_2_ group for the -CN group, on average 2 kcal/mol are lost in the V_s, M_^II^. From -CN to -F, 13 kcal/mol are lost; 7 kcal/mol from -F to -H; 2 kcal/mol from -H to -NH_2_; 1 kcal/mol from -NH_2_ to -tBu; and 2 kcal/mol from -tBu to -N(CH_3_)_2_. Moreover, from these results, it can be proposed that the metal center is much more electrostatically sensible to the presence of electron-withdrawing groups than to electron-donating groups.

### 2.2. Dimeric Complexes

After the prior MEP analysis, two sets of dimeric complexes were proposed, and their optimizations served for posterior analysis. These two sets of dimers were designed for studying and further drawing some conclusions on the capabilities of the metal centers to act as electron acceptors or electron donors. Therefore, the two sets of complexes used either pyridine (py, C_5_H_5_N) or pentafluoroiodobenzene (C_6_F_5_I) acting as the electron-donating/accepting counterpart, respectively. The pyridine complexes were designed by situating the pyridine ring above the metal center, with the nitrogen lone pair (LP) pointing towards the metal atom. The pentafluoroiodobenzene complexes were oriented in the same manner, exploiting the σ-hole generated across the C-I bond, thus having the organic ring situated over the metal center, along the C-I bond axis. The electrostatic potential at these locations (LP of py and σ-hole of C_6_F_5_I) coincides with the minimum (most negative) for the LP of the N in pyridine and with the maximum (most positive) for the σ-hole of the C-I bond of the pentafluoroiodobenzene, as can be observed in Table 1 and Figure 1. For both sets, the rest of the rings are pointing outside the molecular plane of the previously studied monomers, having them perpendicular to the molecular plane in an attempt to have the minimum ancillary interactions. The resulting interaction energies of the dimers would illustrate how the different substituents aid or harm the ability of each metal center to act as an electron donor or acceptor. Larger negative values would mean that the ability of the metal center to act as either donor or acceptor is being enhanced. Thus, one would expect, based on the previous MEP analysis, that the metal acceptors would be better as electron donors (and thus display a larger interaction energy with the pentafluoroiodobenzene molecule) with the corresponding electron-donating substituents which give a more negative electrostatic potential over the metal. In the case of the pyridine dimers, the opposite would be expected; larger interaction energy values would ideally be found in the electron-accepting substituents that lower electrostatic potential values over the metal center. Figure 2 serves as an example of the design of the dimeric systems.

In all cases, a global minimum was achieved as a result of the optimization, yielding favorable interaction energies. The interaction energies and M-N/M-I distances are indicated in Table 2 for the pyridine set of dimers and in Table 3 for the C_6_F_5_I set of dimers.

In a similar fashion as was done for the MEP analysis, some conclusions can be drawn when comparing fixed variables. In all cases, negative, favorable interaction energies were achieved, ranging from −10.3 kcal/mol for the most stable **4A:py** complex (M = Ni^II^, Y = O and X = -NO_2_) to −2.2 kcal/mol for the least stable dimers of **20A** and **20B:py** (M = Pt^II^, Y = O, S and X = -N(CH_3_)_2_). The results obtained agreed with the results of the MEP analysis. The metal centers acting in an electron-accepting role meant that the monomers with the more positive electrostatic potential over the metal would be better at interacting with the negative lone pair of pyridine’s nitrogen atom. Therefore, when augmenting the nucleophilicity of the metal—in other words, when descending in group 10 of the elements—worse interaction energies and greater distances were obtained.

Interestingly, however, in some cases, when having sulfur atoms as the chalcogen moiety, the interaction energies were worse than those having oxygen atoms as the chalcogen metal-chelating moiety, in disagreement with the results that would be expected by simply analyzing the electrostatic potential over the metal cation for the monomeric parts. This could mean that part of the interaction between the monomers and the pyridine ring would be also affected by the electronic cloud of the sulfur atoms and thus repelling the pyridine ring. This situation can be noted for the dimers where Ni^II^ and Pd^II^ acted as metal centers and seems to be reverted in certain dimers when Pt^II^ is acting as the metal center (16A-B, 19A-B, 20A-B, 21A-B: py dimers). In these specific instances, the X group coincidentally exhibited electron-donating characteristics. This observation underscores the significant influence of long-range effects, in conjunction with the chalcogen moiety, on the metal’s capability to function as an electron acceptor.

As expected, the X with an electron-donating nature led to worse interaction energies when the metal atom was acting in the role of electron acceptor and better interaction energies when the X group had an electron-withdrawing nature. These results aligned with those expected from the MEP analysis and revealed that the metal centers are more sensitive to the presence of electron-withdrawing X groups than to electron-donating groups.

In the set of dimers where the metal center acted as the electron donor towards the sigma-hole of the C-I bond in pentafluoroiodobenzene (Table 3), larger interatomic distances between the dimers were obtained as well as a smaller extension of interaction energies ranging from −2.4 kcal/mol for dimer **4A:C_6_F_5_I** (M = Ni^II^, Y = O, X = -NO_2_) to −7.4 kcal/mol for complex **20B:C_6_F_5_I** (M = Pt^II^, Y = S, X = -N(CH_3_)_2_). As would be expected, and according with the MEP analysis, the more nucleophilic the metal, the greater the interaction energy. Also, in agreement with the MEP analysis, all the B sets of dimers resulted in greater interaction energies than their A set counterparts. This phenomenon may be attributed to two factors: firstly, the sulfur moiety contributes to the electron donation towards the σ-hole of C_6_F_5_I; secondly, sulfur atoms are not as effective as oxygen atoms in withdrawing electronic density from the metal. As previously mentioned, these factors collectively result in slightly improved interaction energies for the B set of dimers.

Lastly, as expected, the variation of the X group resulted in better interaction energies when it had an electron-donating nature, increasing the nucleophilicity of the metal center.

As a summary, all the dimers studied herein obtained negative attractive interaction energies. Moreover, the interaction energy values depicted the extensive tunability of the interaction strength with the different substituents. As could be expected, heavier electron-withdrawing groups greatly enhanced the electrophilicity of the metal centers, whereas the effect of electron-donating moieties had a more subtle effect on the nucleophilicity of the metal centers. Lower-period metals yielded the most stable pyridine dimers, whereas the opposite was observed for the pentafluoroiodobenzene dimers, where the higher-period metals resulted in the most stable dimers.

#### 2.2.1. Combined Quantum Theory of Atoms in Molecules and Non-Covalent Interaction Plot (AIM/NCIplot) Analyses

For a deeper understanding of the nature of the interactions regarding the metal center and the donor/accepting pyridine/pentafluoroiodobenzene moieties, Bader’s quantum theory of atoms in molecules [31,32] was used conjointly with the non-covalent interaction plot [33] for analysis.

Following the monomeric structures that were selected for the MEP analysis, the corresponding dimeric forms were taken as exemplary models, and they are shown in Figure 2 and Figure 3. The rest are displayed in Appendix A.

In all the pyridine dimers (Figure 2), a bond path interconnecting the metal center and the nitrogen atom was present as well as a bond critical point (CP). The latter was surrounded by a round greenish/blueish RDG surface demonstrating the attractive nature of the interaction, in agreement with the energy results discussed priorly. Also, in the complexes with the smaller N-M^II^ distance, two additional greenish surfaces were present depicting a slight HB interaction between the pyridine C-H and the chalcogen atoms from the metal-containing monomers. These surfaces were most present when the chalcogen atom was sulfur due to its softer, more extended electron cloud. Also, in the shorter-distance dimers containing sulfur atoms, the round RDG surface surrounding the above-mentioned bond CP displayed a somewhat squared shape, showing a slight contamination of the metal–nitrogen interaction by the surrounding sulfur atoms.

In a similar manner, the pentafluoroiodobenzene dimers (Figure 3) showed a bond path interconnecting the metal atoms with the iodine from the C_6_F_5_I molecule and a bond CP surrounded by a greenish/blueish round RDG surface. This time, however, following the trend observed for the interaction energies, the surface increases the blue intensity (more attractive) when descending in the metal group and when the nature of the X group is electron-donating.

Also, it can be noticed that when the chalcogen atom is sulfur, the round RDG surface surrounding the bond CP between the I and M^II^ atoms is square-shaped. These square-shaped surfaces illustrated the contact between the sulfur atoms together with the iodine atom due to the more diffused electron density of all these neighboring atoms adding up and resulting in larger and more extended RDG values, whereas the electron density of the oxygen atoms was not as diffused, resulting in the circular RDG surface present in these dimers, mostly corresponding to the metal—iodine atoms. In very few dimers, two very slim additional surfaces were present, somewhat between the iodine perpendicular electron belts with the π-system of the monomers.

These results indicated the natural attractive interaction of these systems between the metal centers and electron accepting or donating moieties and were well in agreement with the MEP and energetic analyses.

#### 2.2.2. Natural Bond Orbital Analysis

To further characterize the nature of the interactions between the metal centers and the donor/acceptor moieties, natural bond orbital (NBO) analysis was performed [34]. The energetic features and participating orbitals are shown in Table 4 for the pyridine dimers and in Table 5 for the pentafluoroiodobenzene dimers.

The interaction between the metal and the nitrogen atom from the pyridine consists mainly of a lone pair from the nitrogen atom interacting with a lone valence (LV) orbital of the metal (which would be an empty s orbital) for the A set of dimers (Y = O). However, shockingly, this type of interaction was not found in the B set of dimers (Y = S). Stunningly, instead of having one empty s orbital as the acceptor orbital, there were four identical donor/acceptor interactions, being the acceptor orbitals the σ* S-M^II^ orbitals. Following the previous figures, some dimers have been represented in Figure 4 as exemplary models showing these types of interaction.

If we compare the values of the second-order perturbation analysis of the donor → acceptor pairs, we can observe the same trends as in the MEP and the dimer energetic and combined QTAIM/NCIplot analyses. Once more, for these sets of dimers with pyridine as the electron-donating part, the energetic contribution was greater in the A set of dimers than in the B set, and it also increased when the X group was electron-withdrawing, and the metal was heavier in the group 10 elements.

For the dimers where the metal played the electron-donating role, the donor → acceptor orbitals were much more homogeneous. For all dimers, the main donor orbital was a high-energy d orbital from the metal center and the accepting orbital was the σ* C-I. The energetic features are presented in Table 5.

The second-order perturbation analysis donor → acceptor orbital energies followed the expected trend as per the previous analyses. The most energetically favorable interactions were those where the metal center was most nucleophilic, going down group 10 of the elements, and the X group was of an electron-donating nature. However, the B set of dimers (Y = S) did not result in larger energies than the set of A dimers (Y = O). Also, for the lower-period Ni and Pd metals, an additional interaction could be observed where the 4 σ S-M^II^ orbitals would act as donors. This meant that the joint orbital interaction between the dimers would be the added contribution of the high-energy d orbital from the metal and four times the contribution of one σ S-M^II^ → σ* C-I. This event was not observed for the dimers containing Pt^II^. Some exemplary models for these interactions are presented in Figure 5. Also, these σ S-M^II^ → σ* C-I are displayed in Appendix A.

## 3. Materials and Methods

The calculations were performed at the PBE0 [35]-D3 [36]/def2-TZVP [37,38] level of theory. For the heavier elements, the def2-ecp basis set was employed, which takes into consideration the relativistic effects. The use of effective core potentials (ECPs) results in more accurate and less costly results where relativity is not negligible, in the case of atoms with higher nuclear charge [39]. This selection of the functional and basis set was made based on the general applicability, efficiency and reliability of the method, being one of the most generally used, time-economic and robust levels of theory for giving quality results [40]. The monomers and dimers described in this work were optimized using the TURBOMOLE 7.2 software package [41]. The interaction energies were computed by taking the energies of the separate optimized monomers and the supramolecular dimer and calculating the difference between the latter and the prior monomers.

The molecular electrostatic potential surfaces (MEPs) were generated at the same level of theory and visualized using the Gaussview 6.1.1 software [42].

The combined Bader’s quantum theory of atoms in molecules/non-covalent interaction plot (QTAIM/NCIplot) analysis was carried out by means of the Multiwfn 3.8 software [43]. The reduced density gradient (RDG) was plotted at the RDG = 0.5 isovalue, with an electron density cut-off $\rho_{cut-off}$ = 0.05 a.u. and with a color scheme −0.040 ≤ sign (λ_2_)ρ ≤ +0.040 red–yellow–green–blue for depicting strong repulsive–weak repulsive–weak attractive–strong attractive interactions.

The natural bond orbital analysis (NBO) was performed by means of the NBO 7.0 software [44] at the same level of theory.

The visualization and design of the figures were performed using the VMD 1.9.3 software [45].

## 4. Conclusions

The results obtained from this work further corroborate the idea that the group 10 elements can be used as both electron donor or acceptor moieties and that their proficiency as one or the other can be finely tuned with close or even long-range substituents. The described analyses have provided more insight into the non-covalent nature of these interactions and how the most closely positioned substituents can have a direct impact on the metal interaction with exogenous molecules, but also that long-range substituents have a noticeable impact too.

We believe that these results can encourage investigators to pursue the rational use of various substituents which could further improve the interaction of the metals of interest with other exogenous molecules via weak, reversible non-covalent interactions. Thus, this study might be useful for the fields which exploit the use of these metals such as crystal engineering, medicinal chemistry, catalysis and materials science.

## Data Availability

No new data were created or analyzed in this study. Data sharing is not applicable to this article.

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
