# Peer review of "Tuning the Nucleophilicity and Electrophilicity of Group 10 Elements through Substituent Effects: A DFT Study"

_ijms, 2023, doi:10.3390/ijms242115597_

Round 1
Reviewer 1 Report
Comments and Suggestions for Authors
The manuscript entitled "Tuning the nucleophilicity and electrophilicity of Group 10 elements through substituent effects: A DFT study" analysis a series of electron donor (–NH2, –NMe2 and –tBu) and electron withdrawing substituents (–F, –CN and –NO2) were used to tune the nucleophilicity or electrophilicity of a series of square planar Ni2+, Pd2+ and Pt2+ malonate coordination complexes towards a pentafluoroiodobenzene and a pyridine molecule. However, the content innovation and analysis depth are not enough, which needs to be modified before the publication on International Journal of Molecular Science.
Issue 1: The manuscript gives the corresponding software name for drawing, but does not give the corresponding software name for calculation. Please revise it.
Issue 2: The experimental simulation selects a single atom to discuss the reaction between them, but does not discuss the situation of the unit cell, which is of little reference value for the actual experiment, Please provide additional clarification.
Issue 3: The manuscript is mainly based on simulation calculation, no relevant experiments have been done, and the experimental data are all based on the database as a reference, which is not rigorous enough. Please revise it.
Issue 4: In the section 2.1, the results of the action of substituents on the group 10 elements are displayed, but the manuscript does not deeply analyze the bond energy of each bond, so as to discuss the mechanism of action of substituents. Therefore, the reference is still relatively low, please revise it.
Issue 5: In the section 2.2, why can dimeric complexes study and further draw some conclusions on the capabilities of the metal centers to act as electron acceptors or electron donors? Please explain it.
Issue 6: In the section 2.2.1, why squared-shaped RDG surface illustrated the contact between the sulfur atoms altogether with the iodine atom? Please explain it.
Issue 7: The colors of atoms in fig 2 are similar and even confusing. There is less color difference between carbon, hydrogen atoms, and among central atoms. Please revise it.
Comments on the Quality of English LanguageMinor editing of English language required
Author Response
First of all, we would like to thank the this reviewer for his/her careful reading and examination of the manuscript and the suggestions and changes proposed to upgrade the quality of this piece of work. The changes made to the text have been highlighted in yellow; also figures 2 and 3 have been modified and the list of references has been updated. Below we make an issue by issue reply of the changes proposed and question arisen from the reviewer.
Issue 1: The manuscript gives the corresponding software name for drawing, but does not give the corresponding software name for calculation. Please revise it.
Reply: In the Materials & Methods section, the corresponding software for each calculation is depicted, however it is true that the TURBOMOLE 7.2 software package for the calculations was not labelled accordingly as was the rest of the software.
Issue 2: The experimental simulation selects a single atom to discuss the reaction between them, but does not discuss the situation of the unit cell, which is of little reference value for the actual experiment, Please provide additional clarification.
Reply: The experiments herein were artificial and not taken from a real crystal, thus, there were no unit cells. We have added some extra lines in the text to further clarify.
Issue 3: The manuscript is mainly based on simulation calculation, no relevant experiments have been done, and the experimental data are all based on the database as a reference, which is not rigorous enough. Please revise it.
Reply: The referee is right, however the reason why the bibliographic data is not rigorous enough is due to this piece of work being novel in this context.
Issue 4: In the section 2.1, the results of the action of substituents on the group 10 elements are displayed, but the manuscript does not deeply analyze the bond energy of each bond, so as to discuss the mechanism of action of substituents. Therefore, the reference is still relatively low, please revise it.
Reply: The interaction energies obtained were due to the global interaction between both monomers. The dimers were proposed rationally to exploit and further study the interaction between the metal center and one electron donor/acceptor center. It is true that the substituents might harm or aid the interaction directly, however with the posterior QTAIM/NCIplot and NBO analyses little to no direct interaction was observed; meaning that the main effect of the substituents would be to affect the electron donor/acceptor capabilities of the metal center.
Issue 5: In the section 2.2, why can dimeric complexes study and further draw some conclusions on the capabilities of the metal centers to act as electron acceptors or electron donors? Please explain it.
Reply: Done, further explanation has been provided.
Issue 6: In the section 2.2.1, why squared-shaped RDG surface illustrated the contact between the sulfur atoms altogether with the iodine atom? Please explain it.
Reply: Thanks for the suggestion, further explanation has been provided where this comment was made.
Issue 7: The colors of atoms in fig 2 are similar and even confusing. There is less color difference between carbon, hydrogen atoms, and among central atoms. Please revise it.
Reply: In order to improve the quality and clarify any possible confusing atom, the metal centers have been labelled and the X substituent has been depicted.
Reviewer 2 Report
Comments and Suggestions for Authors
The manuscript presents a Density Functional Theory (DFT) study aimed at understanding the tuning of nucleophilicity and electrophilicity of Group 10 elements, with a focus on a series of square planar malonate coordination complexes of Ni2+, Pd2+ and Pt2+. The study utilizes a range of electron-donating and electron-withdrawing substituents, aiming to modulate the reactivity of these complexes towards pentafluoroidobenzene and pyridine molecules. A range of computational techniques have been employed to elucidate the nature and extent of interactions at the molecular level. The work is well-executed, the methods used are sound, and the discussions provide a good insight into the topic of study. However, there are a few areas where minor revisions could further enhance the clarity and impact of this manuscript.
1. It would be helpful to provide more justification for the choice of the PBE0-D3/def2-TZVP level of theory in the context of this study. Have other similar studies used this level of theory, or are there specific benefits in employing this level of theory for the current study?
2. It's appreciable that the authors have considered relativistic effects for heavier elements using the def2-ecp basis set. However, a brief discussion on the importance of accounting for these effects in the context of the studied systems would benefit the readers.
3. The results section is well-structured, but some sub-sections might benefit from a more detailed discussion. For instance, the effects of different substituents on the electrophilicity and nucleophilicity of the complexes could be explored in more depth.
4. It would be advantageous to include comparative analyses with any existing experimental or theoretical data, if available.
5. I noticed that the author used electrostatic potential to conduct a detailed analysis of the system. This is a good tool for analyzing the surface properties of molecules. It is widely used in many fields such as batteries, catalysis, etc., such as Nat. Commun. (10.1038/s41467-023-40969-5); Energy Environ. Sci. (10.1039/d3ee01567j); ACS Appl. Mater. Interfaces (10.1021/acsami.2c14041), which could be mentioned.
Comments on the Quality of English LanguageMinor editing of English language required
Author Response
First of all, we would like to thank the this reviewer for his/her careful reading and examination of the manuscript and the suggestions and changes proposed to upgrade the quality of this piece of work. The changes made to the text have been highlighted in yellow; also figures 2 and 3 have been modified and the list of references has been updated. Below we make an issue by issue reply of the changes proposed and question arisen from the reviewer.
- It would be helpful to provide more justification for the choice of the PBE0-D3/def2-TZVP level of theory in the context of this study. Have other similar studies used this level of theory, or are there specific benefits in employing this level of theory for the current study?
Reply: Done! Described in the Materials & Methods section and the references have been updated.
- It's appreciable that the authors have considered relativistic effects for heavier elements using the def2-ecp basis set. However, a brief discussion on the importance of accounting for these effects in the context of the studied systems would benefit the readers.
Reply: Done! Also incorporated in the Materials & Methods section with an extra reference,
- The results section is well-structured, but some sub-sections might benefit from a more detailed discussion. For instance, the effects of different substituents on the electrophilicity and nucleophilicity of the complexes could be explored in more depth.
Reply: Further discussion has been added after the dimer energies as a summary.
- It would be advantageous to include comparative analyses with any existing experimental or theoretical data, if available.
Reply: There is not much bibliographic data on the matter, however the references on the introduction lead to similar works which can be used as reference.
- I noticed that the author used electrostatic potential to conduct a detailed analysis of the system. This is a good tool for analyzing the surface properties of molecules. It is widely used in many fields such as batteries, catalysis, etc., such as Nat. Commun. (10.1038/s41467-023-40969-5); Energy Environ. Sci. (10.1039/d3ee01567j); ACS Appl. Mater. Interfaces (10.1021/acsami.2c14041), which could be mentioned.
Reply: Good point! Also, the MEP did not have a proper introduction in this work. A small paragraph with the recommended references has been written.
Reviewer 3 Report
Comments and Suggestions for Authors
Manuscript: Tuning the nucleophilicity and electrophilicity of Group 10 ele-2 ments through substituent effects: A DFT study
Recommendation: Minor Revision: may be suitable for publication, but only after substantial changes with valid explanations.
The authors have performed a normal study related to the influence of electron-withdrawing Vs electron donating group to transition metal complex by DFT analysis. Interestingly this includes Bader’s theory of Atoms in Molecules (AIM), Noncovalent Interaction plot (NCI plot), Molecular Electrostatic Potential (MEP) surface, and Natural Bond Orbital (NBO) analyses.
Overall, this paper brings interest to the audience specific to this area. I have some concerns to address before publishing it, it needs a minor revision for the following reasons:
1. The current study is limited to transition metals such as Ni ,Pd and Pt. Any specific reason? please justify the answer.
2. The functional used in the DFT calculations are through PBE, why not try with potential functional such as B3LY or M06?
3. The clarity of Figure 3 can be improve.
Author Response
First of all, we would like to thank the this reviewer for his/her careful reading and examination of the manuscript and the suggestions and changes proposed to upgrade the quality of this piece of work. The changes made to the text have been highlighted in yellow; also figures 2 and 3 have been modified and the list of references has been updated. Below we make an issue by issue reply of the changes proposed and question arisen from the reviewer.
- The current study is limited to transition metals such as Ni ,Pd and Pt. Any specific reason? please justify the answer.
Reply: A small paragraph has been in the introduction addressing this issue. In short, although other groups could also act as both electron donnors/acceptors, we proposed this group as it contains some of the most common donating/accepting elements.
- The functional used in the DFT calculations are through PBE, why not try with potential functional such as B3LY or M06?
Reply: As was suggested by reviewer 2, the Materials and Methods section has been updated to further describe and update the choice of the level of theory in this work. Although some other functionals could better describe the interactions presented herein, we felt that this combination of functional and basis set would give a strong representative set of results as well as being cost-efficient.
- The clarity of Figure 3 can be improve.
Reply: True! Both figures 2 and 3 have been updated as also suggested the first reviewer,